# Promising Energetic Polymers from Nanostructured Bacterial Cellulose

**DOI:** 10.3390/polym15092213

**Published:** 2023-05-07

**Authors:** Yulia A. Gismatulina

**Affiliations:** Bioconversion Laboratory, Institute for Problems of Chemical and Energetic Technologies, Siberian Branch of the Russian Academy of Sciences (IPCET SB RAS), Biysk 659322, Russia; julja.gismatulina@rambler.ru

**Keywords:** nanostructured bacterial cellulose, nitration, stabilization, nanostructured bacterial cellulose nitrates, nitrogen content, energetic polymers

## Abstract

This study investigated the nitration of nanostructured bacterial cellulose (NBC). The NBC, obtained using symbiotic *Medusomyces gisevii* Sa-12 as the microbial producer and then freeze-dried, was nitrated herein by two methods, the first using mixed sulphuric–nitric acids (MA) and the second using concentrated nitric acid in the presence of methylene chloride (NA+MC). The synthesized samples of NBC nitrates (NBCNs) exhibited 11.77–12.27% nitrogen content, a viscosity of 1086 mPa·s or higher, 0.7–14.5% solubility in an alcohol–ester mixture, and 0.002% ash. Scanning electron microscopy showed that the nitration compacted the NBC structure, with the original reticulate pattern of the structure being preserved in full. Infrared spectroscopy for the presence of functional nitro groups at 1658–1659, 1280, 838–840, 749–751 and 693–694 cm^−1^ confirmed the synthesis of cellulose nitrates in particular. Thermogravimetric and differential thermal analyses showed the resultant NBCNs to have a high purity and high specific heats of decomposition of 6.94–7.08 kJ/g. The NBCN samples differ conceptually from plant-based cellulose nitrates by having a viscosity above 1086 mPa·s and a unique 3D reticulate structure that is retained during the nitration. The findings suggest that the NBCNs can be considered for use in novel high-tech materials and science-driven fields distinct from the application fields of plant-based cellulose nitrates. The NBCN sample obtained with NA+MC has the ability to generate an organogel when it is dissolved in acetone. Because of the said property, this NBCN sample can find use as a classical adhesive scaffold and an energetic gel matrix for creating promising energetic polymers.

## 1. Introduction

Energetic polymers are compounds that comprise energetic (explosophoric) groups [1] such as nitro-, azido- and other groups, and their burning products contain a large amount of nitrogen gas. Because of their unique combination of physicochemical and process- and performance-related characteristics, cellulose nitrates are in-demand energetic polymers [2]. Cellulose nitrates are nowadays expanding their application fields in many industries, including those related to the manufacture of membranes, dyes, lacquers, filters, biosensors, rocket propellants, explosives, and energetic materials [3,4,5].

Cellulose nitrates are generally produced by homogeneous or heterogeneous processes and from numerous cellulose sources such as cotton and wood and the ever-expanding range of alternative plant-based feedstocks [1,6,7,8,9,10,11,12,13,14,15,16]. However, a problem associated with plant raw materials is that these contain, in addition to cellulose, substances such as lignin, hemicelluloses and minerals that require removal before the synthesis of cellulose nitrates, thereby causing damage to the environment [17]. A promising feedstock for cellulose nitrate synthesis is nanostructured bacterial cellulose (NBC), which demonstrates a number of unique properties when contrasted with plant cellulose. NBC is produced as highly hydrated membranes containing no non-cellulosic constituents and exhibits a higher molecular weight and crystallinity index of the ultrafine reticulate structure [18,19,20,21]. In contrast with plant-based cellulose, which exhibits a crystallinity of 40–85%, the crystallinity of NBC can exceed 80% [18] or 90% [21]. The degree of polymerization of plant-based cellulose ranges typically from 2000 to 6000 [19], while that for NBC is much higher, reaching 14,000.

The conceptual feasibility of synthesizing cellulose nitrates from NBC was first demonstrated by treating NBC with concentrated HNO_3_ in methylene chloride at 4 °C [22]. However, due to the extremely small quantity of the resultant NBC sample, the authors of the relevant study failed to determine its characteristics, with the exception of its nitrogen content. One of the more recent and extensive works on the preparation of NBC nitrates (NBCNs) was carried out by Sun et al. [23] who examined in more detail how different nitration parameters (mass ratio of mixed sulphuric–nitric acids, temperature, time, and mass ratio of substrate to nitrating agent) influenced the degree of substitution of NBCNs. However, no data were reported on viscosity and solubility, which are essential characteristics for the practical use of NBCNs and are determinant of the processing and physicochemical properties of ready-made products. Luo et al. [24] synthesized NBCNs from NBC over a wide range of nitrogen content, and NBCNs exhibited a rigid molecular chain in a dilute acetone solution. A more recent achievement in the art of NBCN synthesis was reported by Roslan et al. [25] whose X-ray diffraction analysis revealed that the treatment of NBC with the nitrating mixture amorphized the crystalline structure. Jamal et al. [26] carried out a comparative synthesis of cellulose nitrates from the two raw sources: plant-based (*Ceiba pentandra* L. tree fluff) and microbial (NBC from *Nata de Coco*). The cellulose nitrates from both sources were found to have a high degree of substitution, with the NBCN showing superiority.

The broad application prospects and advantages of nanostructured energetic composites have been described in a review paper [27] as far back as 2014. Research focused on finding new energetic binders with a high purity and a unique network structure, particularly those based on NBC, has now reached its highest peak of demand [28,29,30,31,32,33,34,35,36]. NBCN is being studied as the basic component of gun propellant because it features an ultrafine, high-purity fibrous network and a more stable structure compared with plant-based cellulose nitrates [28,29]. Another equally important science-driven application field for cellulose nitrates, including NBC, is the fabrication of items such as ionizing-radiation detectors, biological indicators, biosensors, chips, semi-permeable membranes, selective sorbents, and adhesives for electronic applications [3,34,37,38].

Thus, studies focused on the synthesis of nanostructured cellulose nitrates are of high importance. We have previously demonstrated [39,40] a conceptual feasibility of synthesizing cellulose nitrates by treating NBC samples with industrial mixed sulphuric–nitric acids (MA) in a 1:50 mass ratio of substrate to nitrating agent. In the present study, the substrate used was NBC, derived from *Medusomyces gisevii Sa-12* on a desired scale [41]. The differences from our previous studies are, first, the method of sample preparation for the nitration and, second, the nitration conditions and methods, which may enhance the nitrogen content of the target product—cellulose nitrates. The present work examined two nitration methods: (i) the traditional method using mixed acid and (ii) another method using concentrated nitric acid in the presence of methylene chloride. The first method was chosen because it is commonly used in industry [42]. The second was chosen because organic solvents, particularly methylene chloride, have the ability to ease the penetration of the nitrating agent into cellulose. This is the first study reporting experimental research in which the two nitration methods of NBC and the behaviour features of NBCNs in acetone are compared.

The present study aimed to comparatively investigate the synthesis of NBCNs by the nitration of the NBC via the two methods described above.

## 2. Materials and Methods

All the reagents and materials used in this study were procured from AO Vekton, Russia.

### 2.1. Substrate for Study

The substrate used in this study was NBC produced by a *Medusomyces gisevii Sa-12* symbiotic culture acquired from the Russian National Collection of Industrial Microorganisms (State Research Institute of Genetics and Selection of Industrial Microorganisms, Moscow, Russia). NBC was biosynthesized on a synthetic nutrient medium under stationary culture in a Millab Binder-400 climatic chamber (Berlin, Germany) at 27 °C under optimum conditions found previously [43]. After the culture was completed, an NBC gel-film was taken out of the nutrient medium surface and washed free of nutrient medium components and cells by stepwise treatment with 2 wt.% NaOH and 0.25 wt.% HCl, this was followed by washing with distilled water until the wash waters became neutral [41]. 

#### 2.1.1. Preparation of NBC for Nitration

The NBC gel-film (Figure 1a) was homogenized to a uniform mass (Figure 1b) in a Midea MC-BL801 blender (Beijiao, China) and then cast into silicone moulds and freeze-dried in an HR7000-M freeze-drier (Harvest Right LLC, North Salt Lake, UT, USA). The freeze-dried NBC (Figure 1c) was ground to fine flakes of 1–3 mm in size (Figure 1d). The freeze-dried NBC prepared for nitration was at most 5% moist.

#### 2.1.2. NBC Quality Measures

The NBC quality was measured by standard analytical methods. The α-cellulose content was determined by the TAPPI standard by treating cellulose with a 17.5% NaOH solution and quantifying the undissolved residue after being washed with a 9.5% NaOH solution and water, followed by drying [44]. The acid-insoluble lignin content was determined by the TAPPI standard using 72% H_2_SO_4_ [45]. Pentosans were transformed in boiling 13 wt.% HCl solution into furfural, which was collected in the distillate and determined on a xylose-calibrated UNICO UV-2804 spectrophotometer (United Products and Instruments, Dayton, NJ, USA) at a wavelength of 630 nm using the orcinol-ferric chloride reagent [46]. The ash content was determined by the TAPPI standard [47]. The degree of polymerization (DP) was determined by the outflow time of cellulose solution in cadoxene (cadmium oxide in ethylenediamine) from a VPZh-3 viscometer (Ecroskhim Ltd., Saint-Petersburg, Russia) with a capillary diameter of 0.92 mm [48]. The moisture was measured on an Ohaus MB23 moisture analyser (Parsippany, NJ, USA).

All experiments were completed in triplicate and data are expressed as average values.

#### 2.1.3. Structural Analysis and Coupled TGA/DTA of NBC

The surface morphology of the NBC fibres was examined by scanning electron microscopy (SEM) on a Jeol GSM-840 electron microscope (Tokyo, Japan) after sputter-coating a Pt layer of 1–5 nm thick. 

IR spectra of NBC were obtained on an Infralum FT-801 spectrometer (OOO NPF Lumex-Sibir, Saint-Petersburg, Russia) operating at 4000–500 cm^−1^. For IR spectroscopy, NBC was pressed into pellets with potassium bromide in an NBC:KBr ratio of 1:150.

The coupled thermogravimetric and differential thermal analyses (TGA/DTA) were performed on a Shimadzu TGA/DTG-60 thermal analyser (Nakagyo-ku, Japan) under the following conditions: 0.5 g sample weight, 10 °C/min heating rate, 350 °C maximum temperature, and nitrogen as the inert environment.

### 2.2. Nitration of NBC

#### 2.2.1. Nitration of NBC with Mixed Sulphuric–Nitric Acids (MA) and Stabilization 

A weighed portion of NBC (7 g) was treated with commercial mixed sulphuric–nitric acids (MA) that had a 14% water content. The mass ratio of NBC to MA was 1:160, temperature was 25–30 °C, and nitration time was 40 min. After the nitration was completed, the NBCN MA sample was washed and subjected to high-temperature stabilization as follows: treatment with water for 1 h at 85–95 °C, then treatment with a 0.03% sodium carbonate solution for 3 h at 85–95 °C, and then again with water for 1 h at 85–95 °C.

#### 2.2.2. Nitration of NBC with Concentrated Nitric Acid in Methylene Chloride (NA+MC) and Stabilization 

A weighed portion of NBC (7 g) was treated with freshly distilled concentrated nitric acid (99%) in the presence of methylene chloride (NA+MC) in a ratio of 20:80. Prior to the process, the NBC was pre-wetted with half of the methylene chloride mass, afterwards the wetted sample was submerged into the working acid mixture prepared from the remainder of methylene chloride and nitric acid. The mass ratio of NBC to NA+MC was 1:90, temperature was 25–30 °C, and nitration time was 30 min. Upon completion, the NBCN NA+MC sample was successively washed with methylene chloride and ethanol. The high-temperature stabilization was performed with water for 1 h at 85–95 °C.

#### 2.2.3. Calculation of NBCN Yield

The yield of NBCN (%) was calculated by Formula (1): W = (m × 100)/m_initial_(1)
where W is the yield of NBCN (%), m is the weight of the NBCN sample (g) and m_initial_ is the initial weight of the NBC sample (g).

#### 2.2.4. Analysis of NBCN

Prior to analysis, the NBCN samples were dried for 1 h at 100 °C.

The nitrogen content was quantified by the ferrous sulphate method [49,50,51] by which NBCN is saponificated with concentrated (conc.) sulphuric acid, and the formed nitric acid is reduced with iron (II) sulphate to nitrogen oxide that generates, in excess of iron (II) sulphate, a [Fe(NO)]SO_4_ complex compound that colours the solution yellow-pink. The nitrogen content of the original cellulose, as determined by the ferrous sulphate method, was equal to zero.

The NBCN viscosity was determined by measuring the flow time of a 2% NBCN solution in acetone out of a VPZh-1 capillary column (OOO Ecohim, Saint-Petersburg, Russia). The NBCN (1 g) solubility in the alcohol–ester mixture (150 ml) at an alcohol-to-ester ratio of 1 to 2 was measured by filtering the NBCN residue insoluble in the alcohol–ester mixture, followed by drying and weighing. The ash content was quantified by slowly decomposing NBCN with conc. HNO_3_ upon heating, followed by incinerating and weighing the calcined residue. 

All experiments were completed in triplicate and data are expressed as average values.

#### 2.2.5. Structural Analysis of NBCN

The structures of the NBCN samples were examined by the same methods as was the original NBC (Section 2.1.3).

## 3. Results and Discussion

It was discovered experimentally that the NBC produced by the *Medusomyces gisevii* Sa-12 symbiotic culture contained 99.5% α-cellulose, 0.01% lignin, 0.01% pentosans, 0.01% ash, and had a degree of polymerization of 3600. The obtained data are suggestive of the NBC being highly pure. However, it is well known that not only the purity of the original cellulose, but also its shape and structure, affect the nitration outcomes. The more developed and accessible the cellulose surface, the more uniform the nitration process. Figure 1d displays the shape of the original cellulose that had a well-developed surface, which causes the NBC to be highly reactive. Coupled with the high purity, the well-developed NBC surface holds out hope that the reaction mixture has sufficient access to the entire surface of the cellulose, leading eventually to the synthesis of NBCN samples that have uniform physicochemical properties and morphological structures. Table 1 outlines the basic characteristics of the synthesized NBCN samples. 

The synthesized NBCNs had the following quality measures: 11.77–12.27% N content, 1086 mPa·s or higher viscosity, 0.7–14.5% solubility in mixed alcohol–ester, and 0.002% ash content. The yield of NBCN was 150% for the NBCN MA and 158% for the NBCN NA+MC because the average molecular weight was increased by the substitution of nitro groups for the hydrogen atom in cellulose hydroxyls [42].

The NBCN samples differ considerably between each other by their basic characteristics; more specifically, the NBCN NA+MC exhibited a higher nitrogen content of 12.27% (0.5% higher than that of the NBCN MA) and a higher viscosity—one that could not be determined by the method described in Section 2.2.4, since the dissolution of the weighed portion in acetone produced a thick, transparent acetonogel (Figure 2). The acetonogel was immobile when the beaker was turned upside-down because of the cellulose nitrate molecules being structured in the acetone solution. It is likely that the acetonogel represents a framework composed of the continuous 3D macromolecular network whose voids were populated with low-molecular acetone. The NBCN MA represents a very viscous, totally transparent solution that slowly flows down when the beaker is turned upside-down. It should be noted that both of the NBCN samples were totally soluble in acetone, corroborating the fact that the nitration was uniform and that the synthesized products were cellulose nitrates.

Other studies [32,33,34] have reported a uniform incorporation of energetic compounds into the new 3D reticulate structure, an NBC-based nanogel binder matrix, by using the combined, safe and facile sol-gel and freeze-drying techniques. It can be assumed that, because of its unique structure, the acetonogel we obtained from the NBCN NA+MC can be both a self-contained energetic gel matrix and a combination with other energetic compounds.

The high nitrogen content of the NBCN NA+MC is due to the higher reactivity of the NBC wetted with methylene chloride, as this procedure makes the penetration of the nitrating agent into the original cellulose easier. Because the preparation of the NBCN MA does not include treatment with organic solvents, the aforementioned phenomenon does not occur, and the inner part of the fibres remain inaccessible to the nitronium cation due to the cellulose having a low dielectric constant [52,53]. Compared with the NBCN MA, the NBCN NA+MC has a higher viscosity due to the lack of hydrolytic degradation when nitrated and its mild stabilization, whereas the preparation of the NBCN MA involves mixed sulphuric–nitric acids (MA) containing water and H_2_SO_4_ and prolonged multistep stabilization, leading altogether to a decrease in the NBCN viscosity. The milder conditions of the high-temperature stabilization of the NBCN NA+MC are due to the absence of low-stability sulphuric acid and sulphuric–nitric esters, as well as of the acid that is encapsulated in the fibre and held on the surface [42], which is typical of the NBCN MA.

However, the viscosity of both NBCN samples appreciably exceeds that of commercial, plant-based cellulose nitrates that have a much lower viscosity of 0.6–72 mPa·s [42]. This difference is due to the original NBC having a high degree of polymerization (DP) [54].

The present study has achieved improved nitrogen content values of 11.77–12.27% as compared with our earlier results when nitrating NBC [39,40] where the nitrogen content was 10.96–11.45%. This is because we modified the method of preparing NBC for nitration, which increased the mass ratio of substrate to nitrating agent from 1:50 [39,40] to about 1:90–1:160 and, in aggregate, helped enhance the NBC reactivity towards nitration. 

The difference between the 1:90 and 1:160 mass ratios of substrate to nitrating agent in the present study is associated with the features of the nitration methods chosen. The high ability of nitric acid to penetrate the NBC wetted with methylene chloride allows the use of a lower (1:90) mass ratio of substrate to nitrating agent and, at the same time, achieving the higher-substituted NBCN NA+MC when compared with the NBCN MA prepared at a higher mass ratio of substrate to nitrating agent.

SEM patterns of the original NBC and NBCN samples are displayed in Figure 3, wherefrom it follows that the original NBC retains the reticulate structure in full when nitrated.

It is evident that the original NBC (Figure 3a) represents a reticulate structure of long interlaced fibres with large holes differing in size and shape. Once nitrated, the NBCN structure (Figure 3b,c) became denser and more compact. The nitration partly reorganized the microfibrils—the holes became smaller and acquired a clearer shape, with the reticulate pattern of the structure preserved (Figure 3b,c)—in agreement with the other study results [25].

Figure 4 shows IR spectra of the original NBC and NBCNs derived therefrom. 

The IR spectrum of the original NBC (Figure 4a) has characteristic frequencies of the basic functional groups typical of both plant-based cellulose [14,55] and bacterial cellulose (BC) [25,56], specifically: 3348 cm^−1^, 2895 cm^−1^, 1637 cm^−1^, 1428 cm^−1^, 1163 cm^−1^ and 1111 cm^−1^ which are assigned to the O–H stretching, asymmetric and symmetric stretching of C–H, O–H bending of absorbed water, asymmetric bending vibration of CH_2_, C–O–C stretching, skeletal stretch of C–O, and vibration of the b-glycosidic linkage of cellulose, respectively. 

Following the nitration, the peak intensity of the hydroxyls (3348 cm^−1^ and 2895 cm^−1^ for the NBC; 3576–3579 cm^−1^ and 2916–2921 cm^−1^ for the NBCNs) declined due to the NO_2_ group being partially substituted for the hydroxyl in the nitrocellulose. That said, the more the substitution, the higher the nitrogen content of the NBCN (NBCN NA+MC) and the lower the intensity of the hydroxyls. The presence of the functional groups of nitrocellulose is concurrently confirmed: there appeared two sharp intense peaks observable at about 1658–1659 cm^−1^ and 1280 cm^−1^ corresponding to the asymmetric and symmetric stretching of NO_2_, respectively; there appeared a broader intense peak at 838–840 cm^−1^ relating to the O–NO_2_ stretch; there were less intense band peaks observed at 749–751 cm^−1^ and 693–694 cm^−1^ attributed to the asymmetric and symmetric bending of O–NO_2_, respectively. The identical functional groups are observable in the IR spectra of classical plant-based cellulose nitrates [5,14,15,16] and NBC nitrates [5,23,25,26,29]. Thus, the successful attachment of the NO_2_ group to the cellulose backbone is confirmed [25]. 

Figure 5 shows the TGA/DTA analysis results for the synthesized NBC and NBCN samples. 

Three characteristic TGA regions can be distinguished in the experimental curves of the coupled TGA/DTA (Figure 5a) of the NBC sample: the first region from the temperature of the experiment onset to 120 °C, wherein the sample is dried out to show a 1.35% weight loss and a related endothermic peak; the second region between 120 °C and 410 °C, wherein the sample decomposes with a 91.10% weight loss, accompanied by an endothermic transformation; and the third region between 410 °C and 500 °C, wherein the sample continues to decompose with a minor weight loss of 2.05%. The onset temperature of intensive decomposition was 335.89 °C. It is well known that the higher the onset temperature of decomposition, the greater the thermal stability and the purer the cellulose sample [57]. The obtained data are consistent with the NBC study results reported in [58]. The DTA curve of the NBC sample (Figure 5a) was found to exhibit an endothermic peak at 375.44 °C accompanied by a drop in the NBC sample weight to 86%. In this case, the amount of liberated heat was 4.21 kJ/g. The 86% weight loss is indicative of the NBC sample purity being much higher than that of plant-based cellulose; for instance, the giant reed-derived cellulose is 83% pure, which is due to the giant reed cellulose having less α-cellulose, 91.8% versus 99.5% in the NBC [59].

It was discovered from the TGA results for the NBCN samples (Figure 5b,c) that the major thermal degradation of the NBCN samples commences at 201.53–203.79 °C and continues to 270 °C, the weight of the samples changing by 80.65–84.25%. The NBCN samples further continue to decompose with a slight weight loss of 7.68–9.77%. The DTA curves of the NBCN samples (Figure 5b,c) were found to exhibit one narrow exothermic peak at 212–214 °C accompanied by a drop in the NBCN sample weight to 81–84%, suggestive of the high chemical purity of the synthesized NBCN samples. The onset temperature of intensive decomposition of the synthesized NBCN samples was 201–204 °C. The obtained results are on a par with the data on cellulose nitrates derived from commercial cotton cellulose [5], microcrystalline cellulose [14] and NBC [23]. By comparing the DTA curves of the NBCN samples and the original NBC, it is clear that the exothermic peak temperature of the nitrated cellulose diminishes from 375.44 °C to 212–214 °C, which is attributed to the homolytic cleavage of the thermally instable O–NO_2_ bonds that speed up the thermal degradation of the polymeric cellulose chains. The synthesized NBCN samples feature high specific heats of decomposition of 6.94–7.08 kJ/g—in support of their energy capacity.

## 4. Conclusions

Nanostructured bacterial cellulose (NBC) was obtained using symbiotic *Medusomyces gisevii* Sa-12 as the microbial producer. This was freeze-dried, homogenized in a blender, and nitrated herein by two methods: one using mixed sulphuric–nitric acids and another utilizing concentrated nitric acid in the presence of methylene chloride.

SEM microscopy showed that the nitration resulted in a more compact NBC structure, with the original reticulate pattern of the structure being preserved in full. IR spectroscopy detected the basic functional groups of cellulose in the original NBC sample, and the basic functional nitro groups in both of the NBCNs. TGA/DTA analyses showed that the synthesized NBCN samples have a high purity and a high specific heat of decomposition—6.94–7.08 kJ/g—corroborating their energy capacity. The synthesized NBCNs can be considered for use as new, high-tech alternatives to plant-based cellulose nitrate materials in science-driven fields. The NBCN NA+MC sample differs from the NBCN MA in that it has a higher nitrogen content (0.5% higher) and generates an acetonogel when it is dissolved in acetone. Due to this property, the NBCN NA+MC sample can be used as a classical adhesive scaffold or an energetic gel matrix to design energetic polymers.

## Figures and Tables

**Figure 1 polymers-15-02213-f001:**
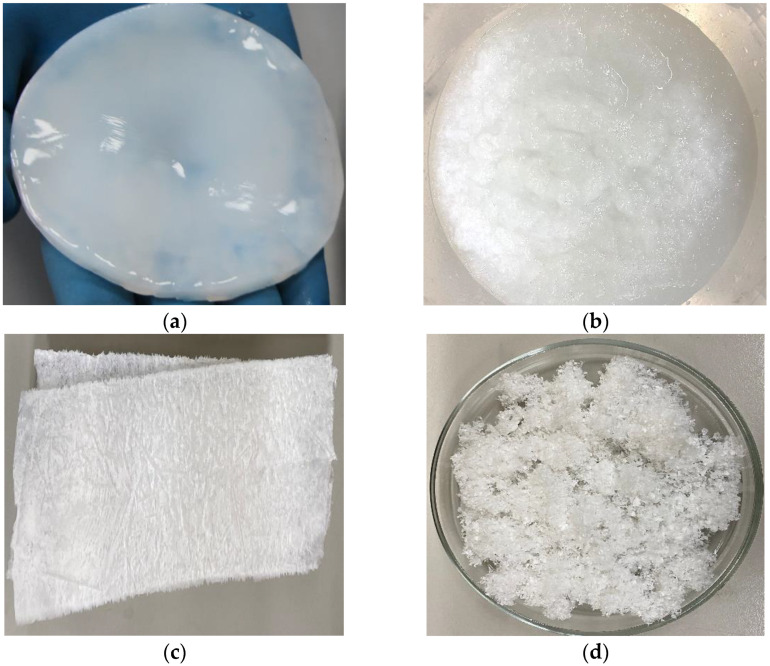
Visualization of the preparation of NBC for nitration: (**a**) NBC gel-film; (**b**) homogenized, wet NBC mass; (**c**) freeze-dried NBC; and (**d**) freeze-dried, ground NBC.

**Figure 2 polymers-15-02213-f002:**
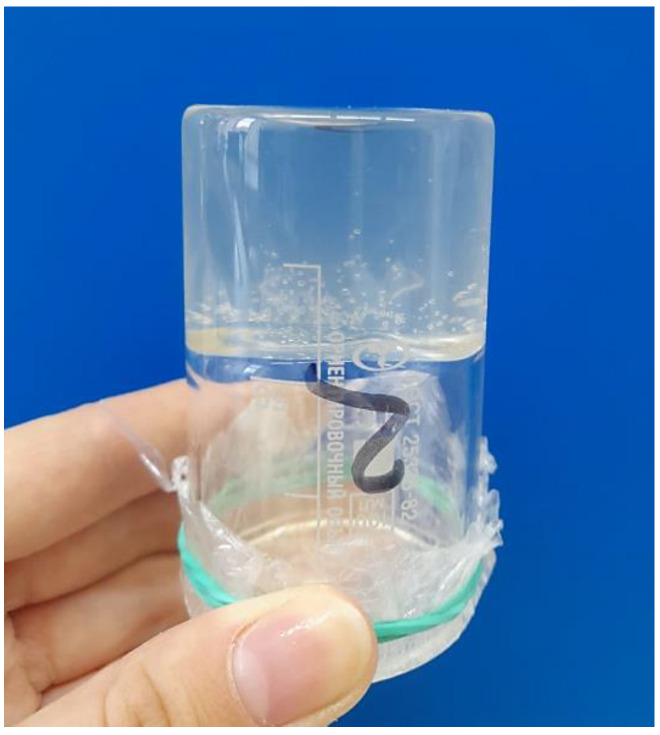
Acetonogel from NBCN NA+MC in acetone solution, the beaker turned upside-down.

**Figure 3 polymers-15-02213-f003:**
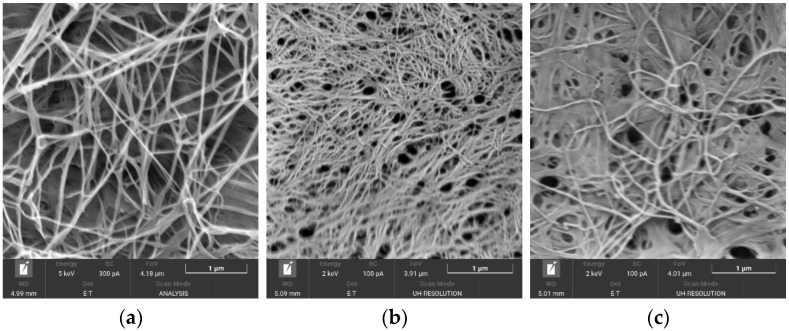
SEM images: (**a**) original NBC; (**b**) NBCN MA; and (**c**) NBCN NA+MC.

**Figure 4 polymers-15-02213-f004:**
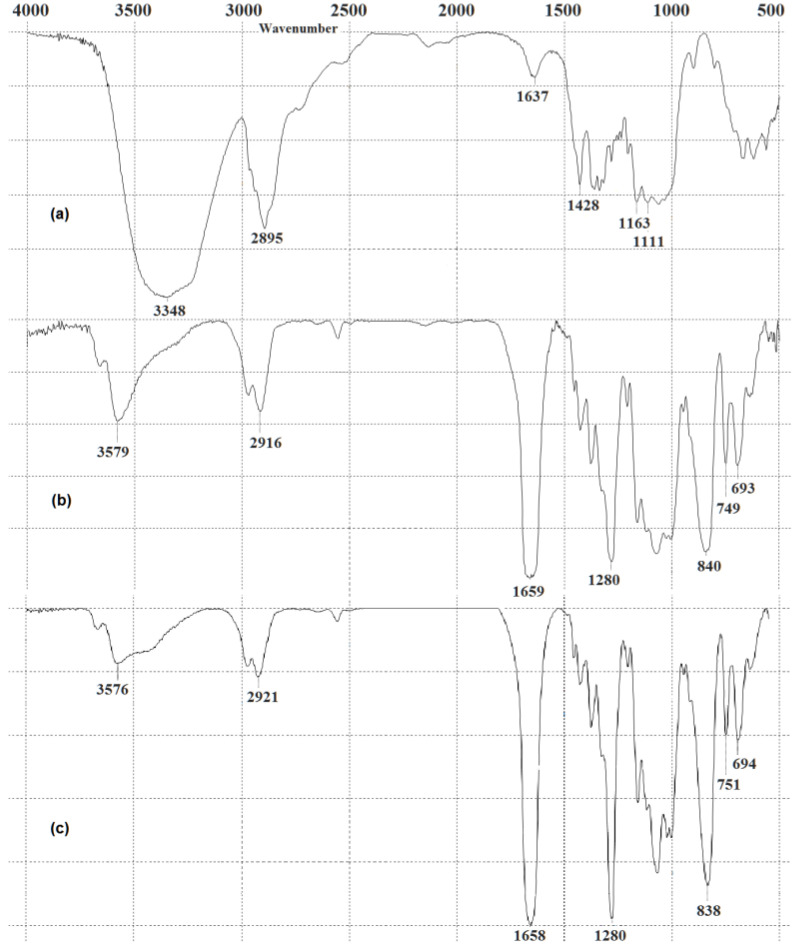
IR spectra: (**a**) original NBC; (**b**) NBCN MA; and (**c**) NBCN NA+MC.

**Figure 5 polymers-15-02213-f005:**
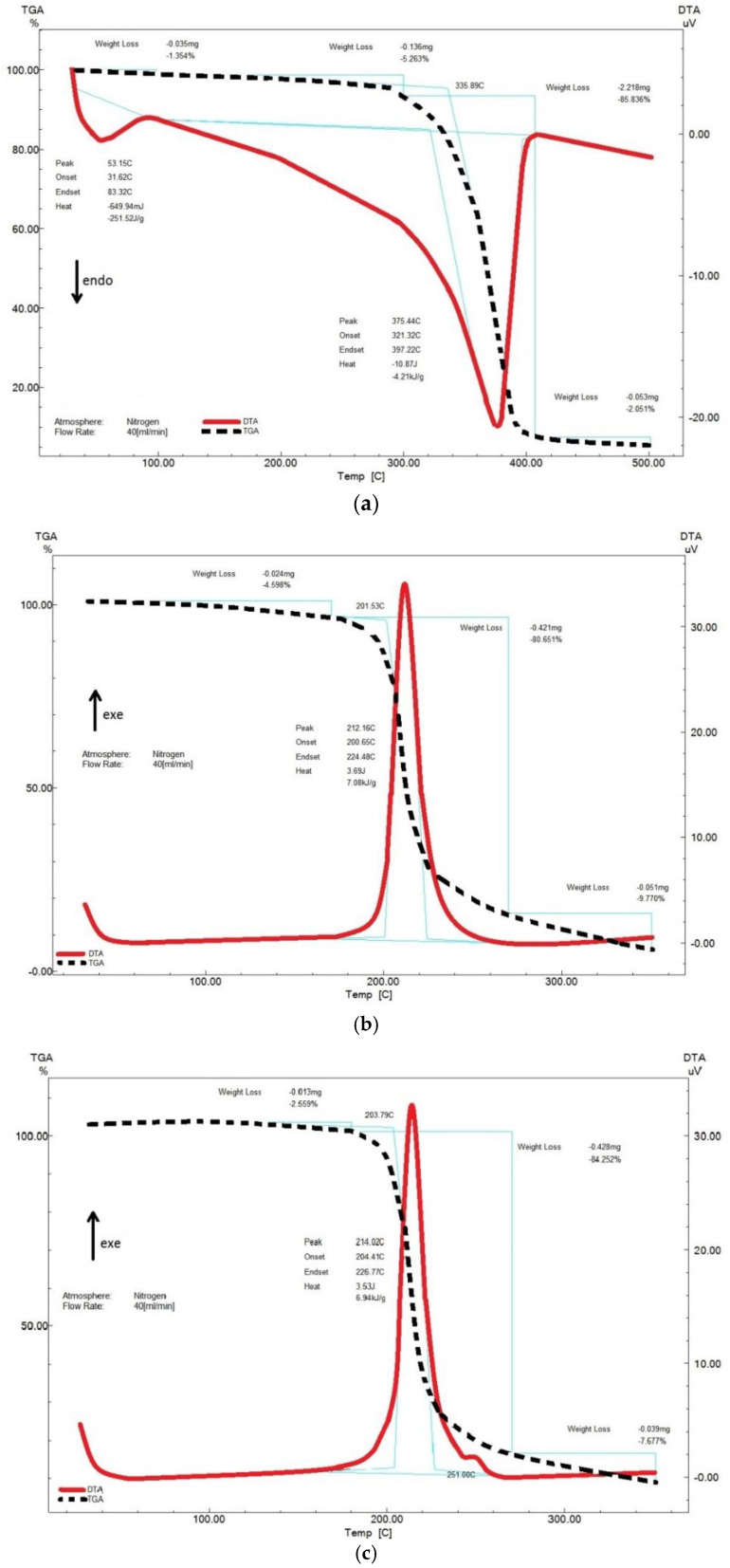
Coupled TGA/DTA images: (**a**) original NBC; (**b**) NBCN MA; and (**c**) NBCN NA+MC. The blue lines are the calculation of the change in the curve region of interest.

**Table 1 polymers-15-02213-t001:** Basic characteristics of synthesized NBCNs.

Sample	N Content, %	Viscosity, 2 % Solution in Acetone, mPa·s	Solubility in Mixed Alcohol–Ester, %	Ash Content, %
NBCN MA	11.77	1086	14.5	0.002
NBCN NA+MC	12.27	acetonogel	0.7	0.002

## Data Availability

The data used to support the findings of this study are included in the article.

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
