# Peer review of "Promising Energetic Polymers from Nanostructured Bacterial Cellulose"

_polymers, 2023, doi:10.3390/polym15092213_

Round 1

Reviewer 1 Report

The manuscript is to investigate the nitration of nanostructured bacterial cellulose (NBC), it is well-written, and sufficiently detailed. I suggest the paper be accepted after minor revisions. The below comments need to be addressed.

1. There are some long sentences that were difficult to understand, for example, in Abstract section, the last sentence, "The NBCN sample ob-22 tained with NA+MC and having the ability to generate an organogel when dissolved in acetone can 23 find its use as a classical adhesive scaffold and an energetic gel matrix for creating advanced en-24 ergetic polymers." need to be revised.

2. In this manuscript, there are many spelling errors, for example, in abstract section, 11.77-12.27% should be replaced by 11.77%-12.27%, 693-694 cm-1 should be replaced by 693 cm-1-694 cm-1, 6.94-7.08 kJ/g should be replaced by 6.94 kJ/g -7.08 kJ/g; Line 152, Page 4, 25-30 ℃ should be replaced by 25 ℃-30 ℃, and so on. the authors should go through the manuscript carefully, and revised and improved.

3. The set in type of the whole manuscript was not beautiful, need to be improved.

4. Only SEM, TGA/DTA, FTIR analysis were performed in this work, but XRD analysis is also a very important tool to characterization cellulose, the author need to add the XRD analysis in this manuscript, that will be better.

4. In Line 310-312, Page 10, the manuscript said that the synthesized NBCN samples showed high specific heats.........in support of their energy capacity. please give the reason of why they can show high specific heats of decomposition, and how they can support their energy capacity.

5. Figure 5, the TGA/DTA analysis image is not clear, the original data and image can not be applied directly here, the author should construct a more clear and normative image using the original data, if you have difficulties, please refer to the figures in other similar manuscript.

6. The conclusion is too lengthy. It needs to be considerably shortened and combined them into one or two paragraphs.

7. The English should be improved. the references should be carefully edited and kept the same format, like Line 424-425, reference 32, The first letter is capitalized, but others are not. the authors should go through the references carefully, and revised and improved.

Author Response

The author's response to the Reviewer has been attached as a separate file.

Reviewer 2 Report

The manuscript "Promising energetic polymers from nanostructured bacterial cellulose" aims to present an improved protocol for bacterial cellulose nitration based on previous studies. The subject is suitable for Polymers Journal, while the manuscript requires some minor details or clarifications. 

First of all, in my opinion, the title sounds too generic, like for a review paper, while the term "energetic" is not extremely relevant for the content of the manuscript, mainly because of the lack of "energetic" experiments, besides DTA.

In the introduction it would be relevant to describe the main criteria and relevant properties of the "energetic materials", and particularly for energetic polymers.

L32: The word "invading" might not be the most appropriate term to describe the context of the study and the sentence could be rephrased.

L39: Ash is not a compound in plants, but a product of burning or pyrolysis, therefore please rephrase. Minerals can be added instead of ash.

L44: Please add a few comparative values for molecular weight and crystallinity,

L84: "use in industry" instead of "use in industrially".

L86: Please rephrase "This is the first study reporting remarkable experimental research" and let the readers to decide how remarkable it is.

L135: It would have been relevant to test different sample weights and heating rates and further correlate them with the heat energy.

L140, L148 and L152: Since only one mass ratio was used for each synthesis, it is hard to have an overview on the nitration process, but it is a start.

L165: The complex is usually written with 5 H2O.

L169-170: The ratio for alcohol-ester mixture and NBCN should also be provided.

Additionally, X-ray diffraction analyses and BET porosity would have been relevant to characterize the samples.

L214: The concept of "energetic gel" should be detailed.

L232-234, 327: The 0.5% improvement does not seems so remarkable. A discussion about the theoretical maximum N content, respectively a yield of cellulose nitration for the performed syntheses, should be provided.

The writing style requires only minor corrections, a few being suggested in the list of comments.

Author Response

(The authors gave the same response as above.)

Reviewer 3 Report

This paper presents the preparation of recticular bacterial cellulose nitrate using mixed sulfuric acid/nitric acid, and nitric acid in methelyene chloride. Results are found as expected. I have some questions and comments as follows:

1.      In the experimental, in case of mixed stabilization after nitration was carried out. Pls write the reason for that in discussion part and why this process was omitted from another experiment (methylene chloride)

2.      Degree of substation (DS) is required by 1H NMR technique to investigate the distribution of nitration scientifically sound. To report the percent nitration doesn’t reflect the DS value. Please be referred to “T. Heinze, T. Liebert, Celluloses and Polyoses/Hemicelluloses, Polymer Science: A Comprehensive Reference Volume 10, 2012, Pages 83-152

3.      TGA/DTA data needs to be redrawn (using raw data is not acceptable).

4.      Thermal analysis needs to be re-written. Note that TGA technique measures % wt loss again heating temperature, DTA using the computer program to follow the RATE of wt loss, at peak it is called Td (degradation temperature). I am curious how authors mentioned “endothermic peak, exothermic peak, heat flow (xx.xx kJ/g) which all of these only get them from DSC analysis but it is not shown this paper.

4.1.  If author insists that TGA/DTA results support the discussion. You MUST provide heat flow thermogram (heating-cooling curve) Y-axis (endothermic arrow up) x-axis (temperature) also provide calculative specific heat capacity equation to support the data shown in TGA/DTA section.

5.      Pls include the reference  Yakaew, P., Phetchara, T., Kampeerapappun, P., Srikulkit, K., Chitosan-Coated Bacterial Cellulose (BC)/Hydrolyzed Collagen Films and Their Ascorbic Acid Loading/Releasing Performance: A Utilization of BC Waste from Kombucha Tea Fermentation, 2022, Polymers 14(21),4544

Author Response

(The authors gave the same response as above.)

Reviewer 4 Report

Comments to Gismatulina

Summary

This study employs a specific symbiotic yeast culture to produce nanostructured bacterial cellulose for subsequent nitration, motivating the exploration with the higher purity and more complex structure of bacterial cellulose in comparison with plant cellulose. Moreover, the study compares two alternative nitration methods assessing the emerging nitrates by means of scanning electron microscopy, infrared spectroscopy as well as thermogravimetric and differential thermal analyses. The authors register the compactification of the cellulose structure, the presence of the characteristic functional nitro groups and similarities in structure between the nitrates corresponding to the different nitration method. However, the study reports differences between the nitrates of the two methods when it comes to viscosity and solvability.

General comments

The study has a relevant topic for the Polymers journal, which also appears to be novel. Furthermore, the manuscript matches the expected structure of a scientific article with the customary sections present. In addition, the authors present sufficient background and experimental detail. Likewise, the presentation of the results is transparent and consistent, though lacking in statistical considerations. As, in the Materials and Methods section the authors inform the reader that all experiments were triplicates, it should be possible to provide some estimate of the statistical variation, helping to decide whether reported differences are significant or if the number of decimals in certain results is justified.

The English language of the manuscript is fine and that also applies to the illustrations, although in some cases the latter could benefit from additional explanation in the text.

Specific comments

Lines 40-41: A promising feedstock for cellulose nitrate synthesis…

Line 53: This point is purely grammatical. The relative clause beginning with that is a non-defining one. (The NBCNs do not have several different viscosities and solubilities, which the clause would be limiting to the essential ones.) Hence, there should be a comma before the clause and the clause should begin with which instead of that.

Line 80: If the methods belong together with nitration, it is clearer to replace the first comma with as well as.

Line 84: common use in industry or common use industrially.

Lin 87: A bit confusing with these acronyms. Was there a definition for BNC? Is it different from NBC?

Lines 89-91: This is almost a repetition of lines 81-83. Maybe better combine the two passages extracting the essential information from both. A scientific text should be rather straightforward.

Line 192: Here and elsewhere it is misleading to represent results as an interval, when there are only two values. Maybe you could use a vertical bar instead.

Line 208: Choice of word.

Lines 232-233: …as compared with…

Line 233: Are these references correct or should it be [39, 40]?

Figure 5: Here a more elaborate explanation would be useful. What are the pale blue lines in the figure? Moreover, why is the exothermic peak in the DTA curve a minimum in subfigure a), whereas the peaks in subfigures b) and c) are maxima.

The English language is generally at an excellent level, but some word choices are a bit unexpected. For example line 208 reports that exactly cellulose nitrates were synthesized, but the word exactly would rather refer to a quantity that can change gradually. You do not throw a dice to observe that the result is exactly three.

Author Response

(The authors gave the same response as above.)

Round 2

Reviewer 3 Report

I couldn't find that authors answer my comments point by point.  Note that TGA disscussion is seriously wrongly written. It is up to author whether or not to reconsider re-writting to get more citation. It is up to you.

Author Response

The response has been uploaded.
